# REPAINT: Knowledge Transfer in Deep Actor-Critic Reinforcement Learning

## Abstract

Accelerating the learning processes for complex tasks by leveraging previously learned tasks has been one of the most challenging problems in reinforcement learning, especially when the similarity between source and target tasks is low or unknown. In this work, we propose a REPresentation-And-INstance Transfer algorithm (REPAINT) for deep actor-critic reinforcement learning paradigm. In representation transfer, we adopt a kickstarted training method using a pre-trained teacher policy by introducing an auxiliary cross-entropy loss. In instance transfer, we develop a sampling approach, i.e., advantage-based experience replay, on transitions collected following the teacher policy, where only the samples with high advantage estimates are retained for policy update. We consider both learning an unseen target task by transferring from previously learned teacher tasks and learning a partially unseen task composed of multiple sub-tasks by transferring from a pre-learned teacher sub-task. In several benchmark experiments, REPAINT significantly reduces the total training time and improves the asymptotic performance compared to training with no prior knowledge and other baselines.

## 1 Introduction

Most reinforcement learning methods train an agent from scratch, typically requiring a huge amount of time and computing resources. Accelerating the learning processes for complex tasks has been one of the most challenging problems in reinforcement learning (Kaelbling et al., 1996; Sutton & Barto, 2018). In the past few years, deep reinforcement learning has become more ubiquitous to solve sequential decision-making problems in many real-world applications, such as game playing (OpenAI et al., 2019; Silver et al., 2016), robotics (Kober et al., 2013; OpenAI et al., 2018), and autonomous driving (Sallab et al., 2017). The computational cost of learning grows as the task complexity increases in the real-world applications. Therefore, it is desirable for a learning algorithm to leverage knowledge acquired in one task to improve performance on other tasks.

Transfer learning has achieved significant success in computer vision, natural language processing, and other knowledge engineering areas (Pan & Yang, 2009). In transfer learning, the teacher (source) and student (target) tasks are not necessarily drawn from the same distribution (Taylor et al., 2008a). The unseen student task may be a simple task which is similar to the previously trained tasks, or a complex task with traits borrowed from significantly different teacher tasks. Despite the prevalence of direct weight transfer, knowledge transfer from previously trained agents for reinforcement learning tasks has not been gaining much attention until recently (Barreto et al., 2019; Ma et al., 2018; Schmitt et al., 2018; Lazaric, 2012; Taylor & Stone, 2009).

In this work, we propose a knowledge transfer algorithm for deep actor-critic reinforcement learning, i.e., REPresentation And INstance Transfer (REPAINT). The algorithm can be categorized as a representation-instance transfer approach. Specifically, in representation transfer, we adopt a kickstarted training method (Schmitt et al., 2018) using a previously trained teacher policy, where the teacher policy is used for computing the auxiliary loss during training. In instance transfer, we develop a new sampling algorithm for the replay buffer collected from the teacher policy, where we only keep the transitions that have advantage estimates greater than a threshold. The experimental results across several transfer learning tasks show that, regardless of the similarity between source and target tasks, by introducing knowledge transfer with REPAINT, the number of training iterations needed by the agent to achieve some reward target can be significantly reduced when compared to

training from scratch and training with only representation transfer or instance transfer. Additionally, the agent's asymptotic performance is also improved by REPAINT in comparison with the baselines.

## 2 RELATED WORK: TRANSFER REINFORCEMENT LEARNING

Transfer learning algorithms in reinforcement learning can be characterized by the definition of transferred knowledge, which contains the *parameters* of the reinforcement learning algorithm, the *representation* of the trained policy, and the *instances* collected from the environment (Lazaric, 2012). When the teacher and student tasks share the same state-action space and they are similar enough (Ferns et al., 2004; Phillips, 2006), *parameter transfer* is the most straightforward approach, namely, one can initialize the policy or value network in the student tasks by that from teacher tasks (Mehta et al., 2008; Rajendran et al., 2015). Parameter transfer with different state-action variables is more complex, where the crucial aspect is to find a suitable mapping from the teacher state-action space to the student state-action space (Gupta et al., 2017; Talvitie & Singh, 2007; Taylor et al., 2008b).

Most of the transfer learning algorithms fall into the category of *representation transfer*, where the reinforcement learning algorithm learns a specific representation of the task or the solution, and the transfer algorithm performs an abstraction process to fit it into the student task. Konidaris et al. (2012) uses the *reward shaping* approach to learn a portable shaping function for knowledge transfer, while some other works use neural networks for feature abstraction (Duan et al., 2016; Parisotto et al., 2015; Zhang et al., 2018). *Policy distillation* (Rusu et al., 2015) or its variants is another popular choice for learning the teacher task representation, where the student policy aims to mimic the behavior of pre-trained teacher policies during its own learning process (Schmitt et al., 2018; Yin & Pan, 2017). Recently, *successor representation* has been widely used in transfer reinforcement learning, in which the rewards are assumed to share some common features, so that the value function can be simply written as a linear combination of the *successor features* (SF) (Barreto et al., 2017; Madarasz & Behrens, 2019). Barreto et al. (2019) extends the method of using SF and generalised policy improvement in Q-learning (Sutton & Barto, 2018) to more general environments. Borsa et al. (2018), Ma et al. (2018), and Schaul et al. (2015a) learn a universal SF approximator for transfer.

The basic idea of *instance transfer* algorithms is that the transfer of teacher samples may improve the learning on student tasks. Lazaric et al. (2008) and Tirinzoni et al. (2018) selectively transfer samples on the basis of the compliance between tasks in a model-free algorithm, while Taylor et al. (2008a) studies how a model-based algorithm can benefit from samples coming from the teacher task.

However, most of the aforementioned algorithms either assume specific forms of reward functions or perform well only when the teacher and student tasks are similar. Additionally, very few algorithms are designated to actor-critic reinforcement learning. In this work, we propose a representation-instance transfer algorithm to handle the generic cases of task similarity, which is also naturally fitted for actor-critic algorithms and can be easily extended to other policy gradient based algorithms.

## 3 BACKGROUND: ACTOR-CRITIC REINFORCEMENT LEARNING

A general reinforcement learning (RL) agent interacting with environment can be modeled in a Markov decision process (MDP), which is defined by a tuple $M = (S, A, p, r, \gamma)$, where $S$ and $A$ are sets of states and actions, respectively. The state transfer function $p(\cdot|s, a)$ maps a state and action pair to a probability distribution over states. $r : S \times A \times S \to \mathbb{R}$ denotes the reward function that determines a reward received by the agent for a transition from $(s, a)$ to $s'$. The discount factor, $\gamma \in [0, 1]$, provides means to obtain a long-term objective. Specifically, the goal of an RL agent is to learn a policy $\pi$ that maps a state to a probability distribution over actions at each time step $t$, so that $a_t \sim \pi(\cdot|s_t)$ maximizes the accumulated discounted return $\sum_{t \geq 0} \gamma^t r(s_t, a_t, s_{t+1})$.

To address this problem, a popular choice to adopt is the model-free actor-critic architecture, e.g., Konda & Tsitsiklis (2000); Degris et al. (2012); Mnih et al. (2016); Schulman et al. (2015a; 2017), where the critic estimates the value function and the actor updates the policy distribution in the direction suggested by the critic. The *state value function* at time $t$ is defined as

$$V^\pi(s) = \mathbb{E}_{a_i \sim \pi(\cdot|s_i)} \left[ \sum_{i \geq t} \gamma^{i-t} r(s_i, a_i, s_{i+1}) \middle| s_t = s \right]. \tag{3.1}$$

Similarly, the *action value function* (also called Q function) is defined by

$$Q^\pi(s, a) = \mathbb{E}_{a_i \sim \pi(\cdot | s_i)} \left[ \sum_{i \geq t} \gamma^{i-t} r(s_i, a_i, s_{i+1}) \middle| s_t = s, a_t = a \right] . \tag{3.2}$$

The actor-critic methods usually rely on the *advantage function*, which is computed as

$$A^\pi(s, a) = Q^\pi(s, a) - V^\pi(s) . \tag{3.3}$$

Intuitively, the advantage can be taken as the extra reward that could be obtained by taking a particular action $a$. The advantage is usually approximated by generalized advantage estimator (GAE) (Schulman et al., 2015b), which is defined as the exponentially-weighted average of the k-step discounted advantage, namely,

$$\hat{A}_t = \hat{A}_t^{\text{GAE}(\gamma, \lambda)} := \sum_{l=0}^{\infty} (\gamma \lambda)^l \left( r_{t+l} + \gamma V^\pi(s_{t+l+1}) - V^\pi(s_{t+l}) \right) , \tag{3.4}$$

where the parameter $0 \leq \lambda \leq 1$ allows a trade-off of the bias and variance.

In deep RL, the critic and actor functions are usually parameterized by neural networks. Then the policy gradient methods can be used to update the actor network. For example, in the clipped proximal policy optimization (Clipped PPO) (Schulman et al., 2017), the policy's objective function is defined to be the minimum between the standard surrogate objective and an $\epsilon$ clipped objective:

$$L_{\text{clip}}(\theta) = \hat{\mathbb{E}}_t \left[ \min \left( r_t(\theta) \cdot \hat{A}_t, \text{clip} \left( r_t(\theta), 1 - \epsilon, 1 + \epsilon \right) \cdot \hat{A}_t \right) \right] , \tag{3.5}$$

where the policy $\pi$ is parameterized by $\theta$, and $r_t(\theta)$ is the likelihood ratio that

$$r_t(\theta) = \frac{\pi_\theta(a_t | s_t)}{\pi_{\theta_{\text{old}}}(a_t | s_t)} . \tag{3.6}$$

Moreover, the clip function truncates $r_t(\theta)$ to the range of $(1 - \epsilon, 1 + \epsilon)$.

## 4 REPRESENTATION AND INSTANCE TRANSFER IN ACTOR-CRITIC RL

In this section, we describe our knowledge transfer algorithm, REPAINT, for actor-critic RL framework. There are two core concepts underlying our approach, i.e., representation transfer and instance transfer. In the representation transfer, we employ a kickstarted training approach (Schmitt et al., 2018) based on the policy distillation. Then, in the next iteration following the kickstarting, we update the student policy using an *advantage-based experience replay*. We do not assume that the teacher and student tasks are similar or drawn from the same distribution.

The REPAINT algorithm is provided in Algorithm 1, where the value function and policy function are parameterized by $\nu$ and $\theta$, respectively. Without loss of generality, we demonstrate the policy update using the Clipped PPO loss stated in equation 3.5, and use a single teacher policy in both representation and instance transfers. In practice, it can be directly applied to any advantage-based policy gradient RL algorithms, and it is straightforward to have different and multiple teacher policies in each transfer step. The more general algorithm and several variants are presented in Section A.

### 4.1 REPRESENTATION TRANSFER: KICKSTARTED TRAINING

In representation transfer, we use a kickstarting training pipeline (Schmitt et al., 2018), which can be viewed as a combination of policy distillation (Rusu et al., 2015) and population based training (Jaderberg et al., 2017). The main idea is to employ an auxiliary loss function which encourages the student policy to be close to the teacher policy on the trajectories sampled by the student. Given a teacher policy $\pi_{\text{teacher}}$, we introduce the auxiliary loss as

$$L_{\text{distill}}(\theta) = H \left( \pi_{\text{teacher}}(a|s) \| \pi_\theta(a|s) \right) , \tag{4.1}$$

where $H(\cdot \| \cdot)$ is the cross-entropy. In order for the agent to maximize its own future rewards, kickstarting adds the above loss to the Clipped PPO objective function, i.e., equation 3.5, weighted at optimization iteration $k$ by the scaling $\beta_k \geq 0$:

$$L_{\text{RL}}^k(\theta) = L_{\text{clip}}(\theta) - \beta_k L_{\text{distill}}(\theta) . \tag{4.2}$$

---

**Algorithm 1** REPAINT algorithm with Clipped PPO

---
**for** iteration $k = 1, 2, \ldots$ **do**
    **if** $k$ is odd **then**
        Collect trajectories $\mathcal{S} = \{(s, a, s', r)\}$ following $\pi_{\theta_{\text{old}}}(s)$
        Fit state value network $V_\nu$ using $\mathcal{S}$ to update $\nu$
        Compute advantage estimates $\hat{A}_1, \ldots, \hat{A}_T$ using equation 3.4
        Perform gradient optimization on $L_{\text{RL}}^k(\theta)$ defined in equation 4.2    // representation transfer
    **else**
        Collect trajectories $\mathcal{S}' = \{(s, a, s')\}$ following $\pi_{\text{teacher}}(s)$        // instance transfer
        Compute $r$ for each transition using current reward function and add to $\mathcal{S}'$
        Compute advantage estimates $\hat{A}'_1, \ldots, \hat{A}'_{T'}$ using equation 3.4
        **for** t=1,\ldots,$T'$ **do**
          **if** $\hat{A}'_t < \zeta$ **then**
            Remove $\hat{A}'_t$ and the corresponding transition $(s_t, a_t, s_{t+1}, r_t)$ from $\mathcal{S}'$
        Perform gradient optimization on $L_{\text{clip}}(\theta)$ defined in equation 3.5

---

In our experiments, the weighting parameter $\beta_k$ is relatively large at early iterations, and vanishes as $k$ increases, which is expected to improve the initial performance of the agent while keeping it focused on the current task in later iterations.

## 4.2 INSTANCE TRANSFER: ADVANTAGE-BASED EXPERIENCE REPLAY

In the instance transfer iteration, we collect training samples following the teacher policy $\pi_{\text{teacher}}$, but compute the rewards using current reward function from the target task. Since the transitions are obtained from a different distribution, we do not update the value network in this iteration. When updating policy network, we prioritize the transitions based on the advantage values and only use the samples that have advantages greater than a given threshold $\zeta$. Moreover, since the teacher policy has been used in collecting roll-outs, we compute the loss without the auxiliary cross-entropy, but replace $\pi_{\theta_{old}}$ with $\pi_{\text{teacher}}$ in equation 3.5.

The idea of prioritizing and filtering transitions is simple but intuitively effective. As mentioned in Section 3, the advantage can be viewed as the extra reward that could be obtained by taking a particular action. Therefore, by retaining only the "good" samples from the teacher policy, the agent can focus on learning useful behavior under current reward function. We note that our filtering approach is related to, but different from, the prioritized experience replay (Schaul et al., 2015b), which prioritizes the transitions in replay buffer by the temporal-difference error (TD error), and utilizes importance sampling for the off-policy evaluation. In comparison, our method uses experience replay in actor-critic framework, where the replay buffer is erased after each training iteration. Unlike the stochastic prioritization which imposes the probability of sampling on each transition, our method directly filters out most of the transitions from replay buffer and equally prioritizes the remaining data, which reduces the training iteration time. In addition, our method performs policy update without importance sampling. All transitions in a batch have the same weight, since the importance can be reflected by their advantage values. The experimental comparison can be found in Section 5.1.

The concept of advantage-based prioritizing has also been used in offline RL most recently (Peng et al., 2019; Siegel et al., 2020; Wang et al., 2020), in order to better leverage the off-policy data. In offline RL, the "advantage weighting" can be regarded as a regularization to either critic or actor, where the model can focus on learning "good" actions while ignoring poor ones. Again, we want to remark that our proposed advantage-based experience replay has a different formulation. To the best of our knowledge, this is the first time that the advantage-based filtering is applied to instance transfer learning. In Figure 2(b), we will compare the performance of several prioritization rules. The discussion is presented in Section 5.1.

We also consider how the sampling experiences from the teacher policy impact the policy gradient update. Our incremental update fits into the framework of the off-policy actor-critic (Degris et al., 2012; Zhang et al., 2019). We ignore the clip function in equation 3.5 for simplicity and replace $\pi_{\theta_{old}}$

with $\pi_{\text{teacher}}$, which leads to

$$\nabla_\theta L_{\text{clip}}(\theta) = \hat{\mathbb{E}}_{s \sim d_{\text{teacher}}, a \sim \pi_{\text{teacher}}} \left[ \rho(s, a) \nabla_\theta \log \pi_\theta(a|s) \hat{A}^{\pi_{\theta_{\text{old}}}} \right] , \qquad (4.3)$$

where $\rho(s, a) = \pi_\theta(a|s)/\pi_{\text{teacher}}(a|s)$, and the advantage estimator is from last kickstarted training iteration. With the new term $\rho(s, a)$, equation 4.3 can be viewed as a weighted sample expectation. Since the samples are drawn from the teacher policy, $\pi_{\text{teacher}}(a|s)$ usually has large values. At the early learning stage, our advantage-based filtering results in a large step size (due to large $\hat{A}^\pi$). Therefore, a small $\rho$ serves as the compensation to the training, which can prevent the system from over-fitting. In later iterations when the student policy is closer to the teacher, the update is close to traditional actor update except the filtering approach, which introduces bias because it changes the distribution in an uncontrolled fashion. In practice, we can adopt REPAINT in the early training stage, and then reduce to traditional actor-critic algorithms. As a consequence, the agent first learns useful teacher behavior to achieve good initial performance, and focuses on the target task afterwards.

### 4.3 DISCUSSION

A critical question in knowledge transfer for RL is how to characterize the *similarity* between source and target tasks, since most of the transfer learning algorithms perform well when the two are similar. If samples from the two MDPs are given, some metrics can be defined to compute or learn the task similarity, e.g., the Kantorovich distance based metric (Song et al., 2016), the restricted Boltzmann machine distance measure (Ammar et al., 2014), the policy overlap (Carroll & Seppi, 2005), and the task compliance defined in Lazaric et al. (2008). However, in general, the similarity is usually unknown before getting any samples, unless other specific information is given. For example, the methods using successor representation (Barreto et al., 2019; Borsa et al., 2018; Schaul et al., 2015a) assume that the reward functions among tasks are a linear combination of some common features, and they only differ in the feature weights, namely, $r(s, a, s') = \sum_i w_i \phi_i(s, a, s')$ with fixed $\phi_i$'s. Then the similarity can be characterized as the distance of the weight vectors.

In this paper, we aim to show that the proposed REPAINT algorithm handles generic cases in task similarity. To demonstrate that, for simplicity, we assume in the experiments that the state and action spaces stay the same between teacher and student tasks, and the reward functions have the form of linear combination of common features. However, we use the cosine distance function to define the task similarity in this paper, namely, the similarity between two tasks with reward functions $r_1(s, a, s') = \phi(s, a, s')^\top w_1$ and $r_2(s, a, s') = \phi(s, a, s')^\top w_2$ can be computed as

$$\text{sim}(r_1, r_2) = \frac{w_1 \cdot w_2}{\|w_1\|\|w_2\|} . \qquad (4.4)$$

We say the two tasks are *similar* if $\text{sim}(r_1, r_2) > 0$. Otherwise, they are considered to be *different*.

In regards to the proposed REPAINT algorithm, we also want to remark that the policy distillation weight $\beta_k$ and advantage filtering threshold $\zeta$ are task specific, which are dependent with the one-step rewards. To this end, one can consider to normalize different reward functions in practice, so that the one-step rewards are in the same scale. In general, larger $\beta_k$ encourages the agent to better match the teacher policy, and larger $\zeta$ leads to that fewer samples are kept for policy update, which in result makes current learning concentrate more on the high-advantage state-action transitions. More investigations on the two parameters can be found in the experiments.

## 5 EXPERIMENTS

In this section, we use two platforms across multiple bench-marking tasks for assessing the REPAINT algorithm (see Figure 1). We first perform experiments on two continuous control tasks, i.e., Ant and Reacher, in MuJoCo simulator (Todorov, 2016). We compare the performance of REPAINT with training from scratch and training with only kickstarting or instance transfer. Note that since the value network cannot be updated in the instance transfer iteration, we implement training with only instance transfer by alternately applying Clipped PPO and instance transfer (namely, setting $\beta_k = 0$ in Algorithm 1). For discrete control tasks, we use the AWS DeepRacer simulator (Balaji et al., 2019) to evaluate different algorithms on complex tasks, such as single-car time-trial race and multi-car racing against bot cars. The detail of our experimental setup is presented in Section B.

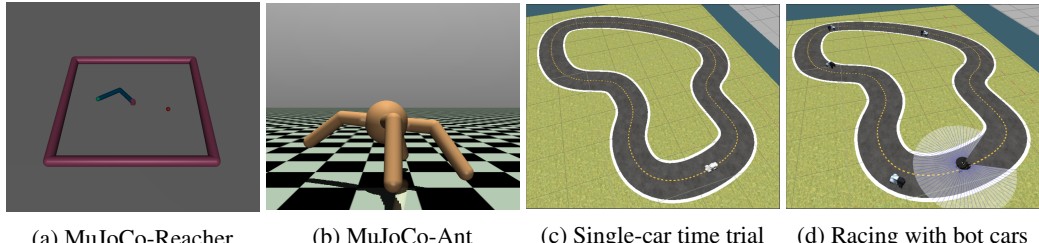

(a) MuJoCo-Reacher     (b) MuJoCo-Ant     (c) Single-car time trial     (d) Racing with bot cars

Figure 1: The simulation environments used in the experiments. Note that the racing car in (c) only has a monocular camera, while in (d) it is equipped by a stereo camera and a Lidar.

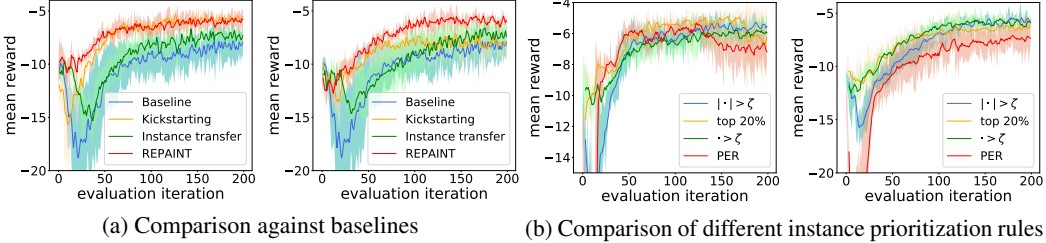

(a) Comparison against baselines      (b) Comparison of different instance prioritization rules

Figure 2: Evaluation performance for MuJoCo-Reacher, averaged across five runs. In each of (a) and (b), we consider both teacher task is similar to (left) and different from (right) the target task.

We consider three different types of task similarities in the experiments based on the cosine similarity (equation 4.4), i.e., the teacher (source) task is a similar task, different task, or a sub-task of the target task (some feature weights are zero in the reward function). We also evaluate the effectiveness of using different teacher policies, cross-entropy weights, and advantage thresholds. Moreover, several prioritization (filtering) approaches in the instance transfer are investigated. More extensive results and discussion on the properties of REPAINT are presented in Section C for completeness.

## 5.1 CONTINUOUS ACTION CONTROL IN MUJOCO PHYSICS SIMULATOR

**MuJoCo-Reacher.** In the target task, the agent is rewarded by getting close to the goal point (distance reward) with less movement (control reward). We first compare our REPAINT algorithm against training with only kickstarting or instance transfer and with no prior knowledge (baseline), based on two different teacher tasks. The first teacher policy is trained with similar reward function but a higher weight on the control reward, where we set it to be 3 as an example, so that the cosine similarity is positive and the learning is transferred from a *similar* task. Another teacher policy is trained in a *different* task, where the agent is penalized when it is close to the goal. In this case, the cosine similarity is zero. After each training iteration, we evaluate the policy for another 20 episodes. The evaluation performance is presented in Figure 2(a). We remark that same trends can be observed from figures of training performance against steps. REPAINT outperforms baseline algorithm and instance transfer in both cases of task similarities, regarding the training convergence time, the asymptotic performance, and the initial performance boosting. Although kickstarted training can improve the initial performance, it has no performance gain in convergence when the teacher behavior is opposed to the expected target behavior. On the other hand, while instance transfer does not boost the initial performance, it surpasses the baseline performance asymptotically in both cases.

We also compare the performance of several advantage prioritization (filtering) rules in the instance transfer of REPAINT, which includes keeping transitions with high absolute values of advantages $(|\cdot| > \zeta)$, keeping top 20% of transitions in the advantage value ranking (top 20%), our proposed filtering rule $(\cdot > \zeta)$, and prioritized experience replay described in Schaul et al. (2015b) (PER). For PER, we used the advantage estimates to compute the prioritization instead of TD errors for a fair comparison, and slightly tuned the $\alpha$ and $\beta$ parameters in the probability computation and importance-sampling weights, respectively. From Figure 2(b), we can observe that the proposed filtering rule and top 20% rule perform better than others on the initial performance, where most of

Table 1: Evaluation performance for MuJoCo-Ant, averaged across three runs. In this experiment, teacher task is similar to the target task.

| Model | Average return score of num_iterations±50 | | | | | | |
|---|---|---|---|---|---|---|---|
| | 100 | 200 | 300 | 400 | 600 | 800 | 1000 |
| Baseline | 37 | 202 | 653 | 1072 | 1958 | 2541 | 3418 |
| Kickstarting (teacher coef.= 3) | 1686 | 2657 | 2446 | 3688 | 4233 | 4744 | 4924 |
| Instance transfer ($\zeta = 1.2$, teacher coef.= 3) | 338 | 1058 | 1618 | 1973 | 3117 | 4163 | 4432 |
| REPAINT ($\zeta = 1.2$, teacher coef.= 3) | 1782 | 2663 | 3150 | 3789 | 4698 | 4996 | 5173 |
| REPAINT ($\zeta = 0$, teacher coef.= 3) | 1747 | 1932 | 2665 | 3245 | 4250 | 4561 | 4809 |
| REPAINT ($\zeta = 0.8$, teacher coef.= 3) | 2004 | 2593 | 2987 | 3638 | 4299 | 4847 | 5074 |
| REPAINT ($\zeta = 1.5$, teacher coef.= 3) | 2009 | 2605 | 2764 | 3309 | 4061 | 4838 | 5116 |
| REPAINT ($\zeta = 1.2$, teacher coef.= 5) | 1034 | 2515 | 2998 | 3492 | 4621 | 4855 | 5028 |
| REPAINT ($\zeta = 1.2$, teacher coef.= 10) | 267 | 1423 | 1785 | 2019 | 2504 | 2914 | 3420 |

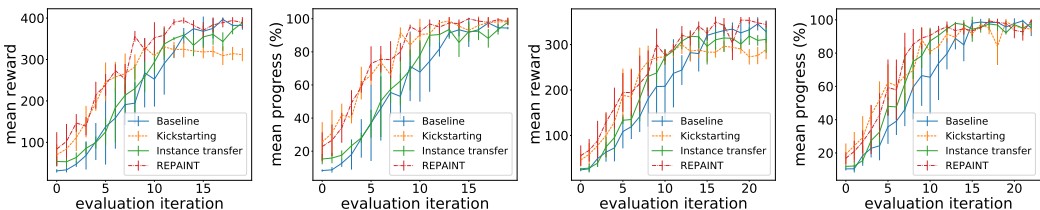

(a) Outer-lane reward task with inner-lane teacher      (b) Inner-lane reward task with outer-lane teacher

Figure 3: Evaluation performance for DeepRacer single-car time-trial race, including mean accumulated rewards and mean progress (lap completion percentage), averaged across five runs.

the samples are removed for policy update. Moreover, PER does not work as well as other approaches when task similarity is low, since it includes all low-advantage teacher samples which have no merits for the student policy to learn. Therefore, we suggest to use the proposed filtering rule with a threshold $\zeta$ or the ranking-based filtering rule with a percentage threshold in the instance transfer.

**MuJoCo-Ant.** Other than the comparison against baselines, we also assess our REPAINT algorithm with different filtering thresholds here, i.e., the parameter $\zeta$ in Algorithm 1. In this target task, the agent is rewarded by survival and moving forward, and penalized by control and contact cost. The teacher task is a *similar* task, which has the reward function with a higher weight on moving forward. In the experiments, we train each model for 1100 iterations, and evaluate each training iteration for another 5 episodes. The evaluation results are shown in Table 1. We can again observe that, when task similarity is high, training with REPAINT or kickstarting significantly improves the initial performance, reduces the learning cost of achieving a certain performance level, and improves the asymptotic performance. The table also indicates that our algorithm is robust to the threshold parameter. Similar learning progresses are observed from training with different $\zeta$ values. In addition, we evaluate the transfer performance from the source tasks with different levels of cosine similarities. We set the forward coefficients in the reward function of teacher task to be 3, 5, and 10, corresponding to the cosine similarities of 0.87, 0.76, and 0.64. By comparing the performance of REPAINT with $\zeta = 1.2$ and different teacher coefficients, we can see that task similarity impacts the overall training, even when they are all related.

## 5.2 AUTONOMOUS RACING IN AWS DEEPRACER SIMULATOR

**Single-car time trial.** In this experiment, we use two different reward functions, one of which encourages the agent to stick to the inner lane (left of the yellow center-line) and the other rewards the agent when it is in the outer lane. When we use one reward function in the student task, we provide the teacher policy that is trained with the other reward. Therefore, the cosine similarity of teacher and target task is zero. We evaluate the policy for 5 episodes after each iteration. The evaluation performance is presented in Figure 3, where both average return and progress (percentage of a lap the

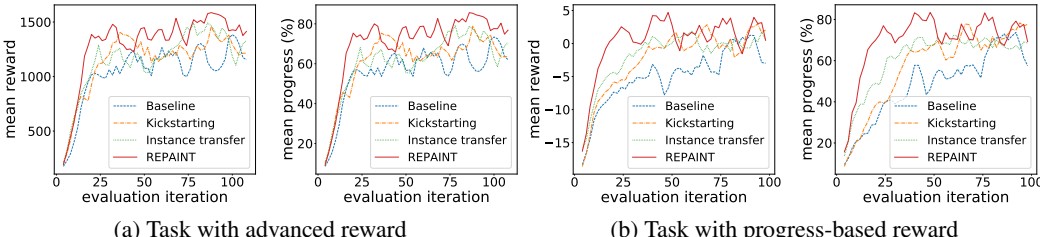

|  | (a) Task with advanced reward | (b) Task with progress-based reward |
|---|---|---|

Figure 4: Evaluation performance for DeepRacer multi-car racing against bot cars, averaged across three runs. The value at each iteration is smoothed by the mean value of nearest 3 iterations.

Table 2: Summary of the experimental results.

| Env. | Teacher type | Target score | $K_{\text{Baseline}}$ | $K_{\text{KS}}$ (pct. reduced) | $K_{\text{IT}}$ (pct. reduced) | $K_{\text{REP}}$ (pct. reduced) | Best scores (KS, IT, REP) |
|---|---|---|---|---|---|---|---|
| Reacher | similar | -7.4 | 173 | 51 (71%) | 97 (44%) | 42 (76%) | -5.3,-5.9,-5.4 |
|  | different |  |  | 73 (58%) | 127 (27%) | 51 (71%) | -6.9,-6.4,-5.2 |
| Ant | similar | 3685 | 997 | 363 (64%) | 623 (38%) | 334 (66%) | 5464,5172,5540 |
| Single-car | different | 394 | 18 | – | – | 13 (28%) | 331,388,396 |
|  | different | 345 | 22 | – | – | 15 (32%) | 300,319,354 |
| Multi-car | sub-task | 1481 | 100 | 34 (66%) | 75 (25%) | 29 (71%) | 1542,1610,1623 |
|  | sub-task | 2.7 | 77 | 66 (14%) | 53 (31%) | 25 (68%) | 4.9,4.2,6.1 |

agent accomplished when it went out of track) are given. Although upon convergence, all models can finish a lap without going off-track, REPAINT and kickstarting again significantly boost the initial performance. However, when the teacher task is very different from the target task, training with kickstarting cannot improve the final performance via transfer. In contrast, instance transfer can still reduce the training convergence time with a final performance better than kickstarting.

In Section C, we also compare the REPAINT algorithm with a commonly used *parameter transfer* approach, i.e., *warm start*, when the task similarity is low. Furthermore, the effect of different cross-entropy weights $\beta_k$ or instance filtering thresholds $\zeta$ is studied, which is presented in Section C.3.

**Racing against bot cars.** The REPAINT algorithm is still helpful when the RL agent needs to learn multiple skills in a task. In the environment of racing against bot cars, the agent has to keep on the track while avoiding slow-moving cars in order to obtain a high return score. Therefore, we first train a teacher policy which is good at object avoidance, namely, the agent is rewarded when it keeps away from all bot cars and gets a penalty when the agent is too close to a bot car and heads towards to it. Then in the target tasks, we use two different reward functions to assess the models. First, we use an *advanced reward* where other than keeping on track and object avoidance, it also penalizes the agent when it detects some bot car from the stereo camera and is in the same lane with the bot. The evaluation performance is shown in Figure 4(a). Since the environment has high randomness, such as agent and bot car initial locations and bot car lane changing, we only report average results across three runs but omit the error bars. One can observe that REPAINT outperforms other baselines regarding the training time needed for some certain performance level and the asymptotic performance. Moreover, another target task with a *progress-based reward* is also investigated, where the agent is only rewarded based on its completion progress, but gets large penalty when it goes off-track or crashes with bot cars. The evaluation performance is provided in Figure 4(b). When the target task is complex and the reward is simple as in this case, it is sometimes difficult for the agent to learn a good policy as it lacks guidance from the reward on its actions. However, a teacher policy can provide guidance for the agent to achieve high accumulated rewards at the early stage of training. From the figure, we can again see that training with REPAINT not only reduces the convergence time, but also largely improves the asymptotic performance compared to other models.

## 6 SUMMARY AND FUTURE WORK

In this work, we have proposed a knowledge transfer algorithm for deep actor-critic reinforcement learning. The REPAINT algorithm performs representation transfer and instance transfer alternately, and uses a simple yet effective method to supplement the kickstarted training, i.e., the advantage-based experience replay. The experiments across several tasks with continuous and discrete state-action spaces have demonstrated that REPAINT significantly reduces the training time needed by an agent to reach a specified performance level, and also improves the asymptotic performance of the agent, when compared to training with no prior knowledge and only kickstarting or instance transfer.

We provide a summary of the transfer performance on the experiments we have conducted, which is given in Table 2. The teacher type indicates whether the teacher task is a sub-task of or similar to the target task, based on the cosine similarity in equation 4.4. The target score is the best return that can be obtained by training from scratch. Then we provide the number of training iterations needed by each model to achieve the target score. The models include training with baseline, kickstarting (KS), instance transfer (IT), and REPAINT (REP). In Section C.5, we also provide the data of wall-clock time. Finally, in the last column, we present the best scores that each knowledge transfer model can achieve in the evaluation. Unlike other baseline algorithms, the superior performance of REPAINT is observed regardless of the task similarity. In contrast, for example, the kickstarted training provides 71% and 58% reduction on MuJoCo-Reacher, when the teacher task is similar versus different. The difference of performance is even larger on DeepRacer multi-car racing when different reward functions are used in the target task. For the DeepRacer single-car time trial, where the teacher task is significantly different from the target task, although the improvement of REPAINT is not as notable as in other tasks, it still outperforms the baseline algorithms. In this case, the kickstarting and instance transfer models are not able to reach the performance level of baseline upon convergence.

In future work, we aim to study how our proposed algorithm can automatically learn the task similarity if unknown, and spontaneously determine the best $\beta_k$ and $\zeta$ values in the training based on the similarity. Our preliminary results in Section C.3 indicate that when the task similarity is low, larger $\beta_k$ values may reduce the asymptotic performance of the agent. Moreover, we are also interested in the dependency of transfer performance on the neural network architectures. We provide some preliminary experimental results in Section C.4.

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

## A  ALGORITHM

In order to directly compare with the baseline algorithms regarding the reduction of number of training iterations, we adopt representation transfer and instance transfer alternately in Algorithm 1, so that the REPAINT performs policy update one time per iteration. Indeed, Algorithm 1 can be easily extended with different alternating ratios other than 1:1 alternating. The corresponding results and discussion can be found in Section C.1. More generically, we can integrate the two transfer steps in one iteration, which is shown in the following integrated REPAINT algorithm with Clipped PPO.

---

**Algorithm 2** Integrated REPAINT algorithm with Clipped PPO

---

**for** iteration $k = 1, 2, \ldots$ **do**
    Collect samples $\mathcal{S} = \{(s, a, s')\}$ by $\pi_{\theta_{\text{old}}}(s)$, and $\mathcal{S}' = \{(s, a, s')\}$ by $\pi_{\text{teacher}}(s)$
    Compute $r$ for each transition using current reward function and add to $\mathcal{S}$ and $\mathcal{S}'$, respectively
    Fit state value network $V_\nu$ using only $\mathcal{S}$ to update $\nu$
    Compute advantage estimates $\hat{A}_1, \ldots, \hat{A}_T$ for $\mathcal{S}$ and $\hat{A}'_1, \ldots, \hat{A}'_{T'}$ for $\mathcal{S}'$ by equation 3.4
    **for** t=1,...,$T'$ **do**
        **if** $\hat{A}'_t < \zeta$ **then**
            Remove $\hat{A}'_t$ and the corresponding transition $(s_t, a_t, s_{t+1}, r_t)$ from $\mathcal{S}'$
    Compute sample gradient of $L^k_{\text{RL}}(\theta)$ defined in equation 4.2 using $\mathcal{S}$
    Compute sample gradient of $L_{\text{clip}}(\theta)$ defined in equation 3.5 using $\mathcal{S}'$
    Update policy network by $\theta \leftarrow \theta + \alpha_1 \nabla_\theta L^k_{\text{RL}}(\theta) + \alpha_2 \nabla_\theta L_{\text{clip}}(\theta)$

---

Note that in Algorithm 2, we can use different learning rates $\alpha_1$ and $\alpha_2$ to control the update from representation transfer and instance transfer, respectively.

So far, we have only used a single teacher policy in the knowledge transfer, and adopt the objective function from only Clipped PPO algorithm. However, it is straightforward to using multiple teacher policies in each transfer step, and our algorithm can be directly applied to any advantage-based policy gradient RL algorithms. Assume there are $m$ previously trained teacher policies $\pi_1, \ldots, \pi_m$. In the instance transfer, we can form the replay buffer $\mathcal{S}'$ by collecting samples from all teacher policies. Then in the representation transfer, the objective function can be written in a more general way:

$$L^k_{\text{RL}}(\theta) = L_{\text{clip}}(\theta) - \sum_{i=1}^{m} \beta_i^k H\left(\pi_i(a|s) \| \pi_\theta(a|s)\right) , \tag{A.1}$$

where we can impose different weighting parameters for different teacher policies.

In addition, the first term in equation A.1, i.e., $L_{\text{clip}}(\theta)$, can be replaced by the objective of other RL algorithms, e.g., A2C (classical gradient policy):

$$L_{\text{A2C}}(\theta) = \hat{\mathbb{E}}_t \left[ \log \pi_\theta(a|s) \hat{A}_t \right] , \tag{A.2}$$

and TRPO (Schulman et al., 2015a):

$$L_{\text{TRPO}}(\theta) = \hat{\mathbb{E}}_t \left[ \frac{\pi_\theta(a|s)}{\pi_{\theta_{\text{old}}}(a|s)} \hat{A}_t - \beta \text{KL}[\pi_{\theta_{\text{old}}}(\cdot|s), \pi_\theta(\cdot|s)] \right] \tag{A.3}$$

for some coefficient $\beta$ of the maximum KL divergence computed over states.

## B  EXPERIMENTAL SETUP

### B.1  ENVIRONMENTS

We now provide the details of our experimental setup. MuJoCo is a well-known physics simulator for evaluating agents on continuous motor control tasks with contact dynamics. In AWS DeepRacer simulator[1], the RL agent, i.e., an autonomous car, learns to drive by interacting with its environment,

---

[1] https://github.com/awslabs/amazon-sagemaker-examples/tree/master/reinforcement_learning/rl_deepracer_robomaker_coach_gazebo

e.g., the track with moving bot cars, by taking an action in a given state to maximize the expected reward. Figures 1(c) and 1(d) present two environmental settings investigated in this work, i.e., single-car time-trial race, where the goal is to finish a lap in the shortest time, and racing against moving bot cars, where four bot cars are generated randomly on the track and the RL agent learns to finish the lap with overtaking bot cars.

In single-car racing, we only install a front-facing camera on the RL agent, which obtains an RGB image with size $120 \times 160 \times 3$. The image is then transformed to gray scale and fed into an input embedder. For simplicity, the input embedder is set to be a three-layer convolutional neural network (CNN) (Goodfellow et al., 2016). For the RL agent in racing against bot cars, we use a stereo camera and a Lidar as sensors. The stereo camera obtains two images simultaneously, transformed to gray scale, and concatenates the two images as the input, which leads to a $120 \times 160 \times 2$ input tensor. The input embedder for stereo camera is also a three-layer CNN by default. The stereo camera is used to detect bot cars in the front of learner car, while the Lidar is used to detect any car behind. The backward-facing Lidar has an angle range of 300 degree and a 64 dimensional signal. Each laser can detect a distance from 12cm to 1 meter. The input embedder for Lidar sensor is set to be a two-layer dense network. In both environments, the output has two heads, V head for state value function output and policy head for the policy function output, each of which is set to be a two-layer dense networks but with different output dimensions. The action space consists of a combination of five different steering angles and two different throttle degrees, which forms a 10-action discrete space. In the evaluation of DeepRacer experiments, the generalization around nearby states and actions is also considered (Balaji et al., 2019), where we add small noises to the observations and actions.

## B.2 HYPERPARAMETERS

We have implemented our algorithms based on Intel Coach[2]. The MuJoCo environments are from OpenAI Gym[3]. If not specified explicitly in the paper, we always use Adam as the optimizer with a learning rate of $3 \times 10^{-4}$, minibatch size as 64, clipping parameter $\epsilon$ as 0.2, $\beta_0 = 0.2$ and $\beta_{k+1} = 0.95\beta_k$ throughout the experiments. The other hyperparameters are presented below.

Table 3: Hyperparameters used in the MuJoCo simulations.

| Hyperparameter | Value |
|---|---|
| Num. of rollout steps | 2048 |
| Num. training epochs | 10 |
| Discount ($\gamma$) | 0.99 |
| GAE parameter ($\lambda$) | 0.95 |
| Beta entropy | 0.0001 |
| Reacher - Advantage Threshold ($\zeta$) | 0.8 |
| Reacher - Num. REPAINT iterations | 15 |
| Ant - Num. REPAINT iterations | 50 |

Table 4: Hyperparameters used in the DeepRacer simulations.

| Hyperparameter | Value |
|---|---|
| Num. of rollout episodes | 20 |
| Num. of rollout episodes when using $\pi_{\text{teacher}}$ | 2 |
| Num. training epochs | 8 |
| Discount ($\gamma$) | 0.999 |
| GAE parameter ($\lambda$) | 0.95 |
| Beta entropy | 0.001 |
| Advantage Threshold ($\zeta$) | 0.2 |
| Single-car - Num. REPAINT iterations | 4 |
| Multi-car - Num. REPAINT iterations | 20 |

---

[2]https://github.com/NervanaSystems/coach
[3]https://gym.openai.com/envs/#mujoco

# C   EXTENSIVE EXPERIMENTAL RESULTS

## C.1   MORE RESULTS ON MUJOCO-REACHER

**Alternating ratios.**   In Algorithm 1, we alternate representation transfer and instance transfer after each iteration. Here, we aim to illustrate the effect of using different alternating ratios by the MuJoCo-Reacher environment. We compare the 1:1 alternating with a 2:1 ratio, namely, two representation transfer (kickstarting) iterations before and after an instance transfer iteration. The evaluation performance is shown in Figure 5. When the teacher task is similar to the target task, adopting more kickstarted training iterations leads to faster convergence, due to the policy distillation term in the loss function. On the other hand, when the task similarity is low, instance transfer contributes more to the knowledge transfer due to the advantage-based experience replay. Therefore, we suggest to set the alternating ratio in Algorithm 1, or the $\alpha_1$ and $\alpha_2$ parameters in Algorithm 2, according to the task similarity between source and target tasks. However, the task similarity is usually unknown in most of the real-world applications, or the similarities are mixed when using multiple teacher policies. It is interesting to automatically learn the task similarity and determine the best ratio/parameters before actually starting the transfer learning. We leave the investigation of this topic as a future work.

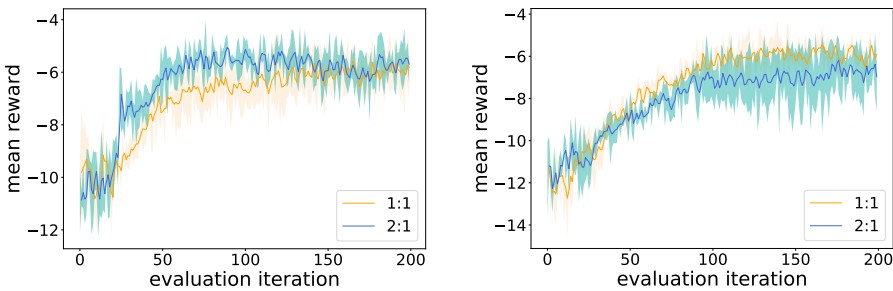

Figure 5: Evaluation performance for MuJoCo-Reacher, averaged across five runs. Left: Teacher task is similar to the target task. Right: Teacher task is different from the target task.

**REPAINT *vs.* Relevance-Based Transfer Learning.**   In order to showcase our contribution of the advantage-based experience replay in the instance transfer, it is also of interest to compare our proposed instance transfer algorithm with other existing methods. In this experiment, we incorporate the kickstarting with the relevance-based instance transfer (Lazaric et al., 2008), and compare its performance with REPAINT in the MuJoCo-Reacher environment. In the relevance-based transfer (RBT), the agent first collects samples from both source and target tasks. For each sample from the source task, RBT computes its relevance to the target samples. Then the source samples are prioritized by the relevance and RBT transfers the samples according to it. Figure 6 shows the evaluation performance of RBT and REPAINT on MuJoCo-Reacher with either a similar source task or a different source task. When a similar task is used in knowledge transfer, as most of other transfer learning methods, RBT works well. However, when the source task is different from the target task, although RBT attempts to transfer the most relevant samples, it has no performance gain over the baseline training. The performance of REPAINT is significantly better than RBT in this case.

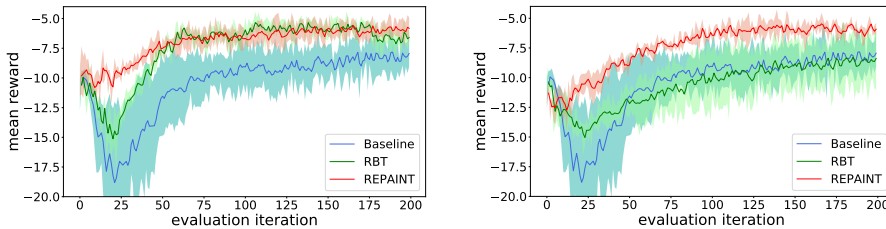

Figure 6: Comparison of relevance-based transfer (RBT) and REPAINT on MuJoCo-Reacher, averaged across five runs. Left: Teacher task is similar to the target task. Right: Teacher task is different from the target task.

## C.2  REPAINT *vs.* WARM-START

We also want to compare our hybrid *representation-instance* transfer algorithm with a widely-used *parameter* transfer approach, i.e., warm-start. In warm-start, the agent initializes with parameters from the teacher policy and conducts the RL algorithm after that. When the target task is similar to the teacher task, it usually works well. But here we compare the two algorithms in the DeepRacer single-car experiment, where the two tasks are significantly different. Figure 7 visualizes the trajectories of the agent on the track during evaluations. Each model is trained for two hours and evaluated for another 20 episodes. From both cases, we can see that although the two reward functions encode totally different behaviors, REPAINT can still focus on current task while learning from the teacher policy. This again indicates the effectiveness of the advantage-based experience replay in the instance transfer. In comparison, training with warm start cannot get rid of the unexpected behavior at convergence due to the reason that it may be stuck at some local optimum. Therefore, initialization with previously trained policies can sometimes jump-start the training with good initial performance, but the method contributes to the final performance only when two tasks are highly related.

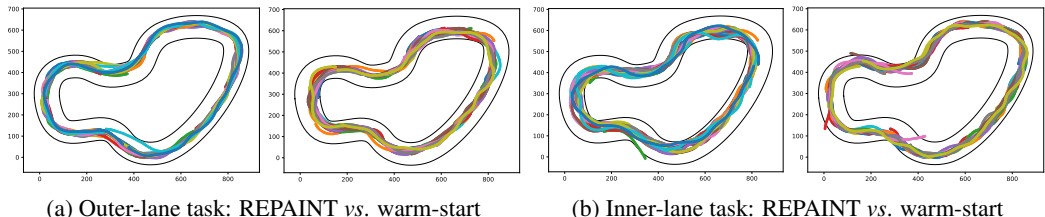

(a) Outer-lane task: REPAINT *vs.* warm-start          (b) Inner-lane task: REPAINT *vs.* warm-start

Figure 7: Trajectories of policy evaluations. In each of (a) and (b), evaluation of the models trained from REPAINT is visualized on the left and that trained with warm-start is on the right.

## C.3  MORE RESULTS ON DEEPRACER SINGLE-CAR TIME TRIAL

In the DeepRacer single-car time-trial task, we also study the effect of different cross-entropy weights $\beta_k$ and instance filtering thresholds $\zeta$, as mentioned in the paper. We first present the results of instance transfer learning with different $\zeta$ values in Figure 8, where we can see that our proposed advantage-based experience replay is robust to the threshold parameter.

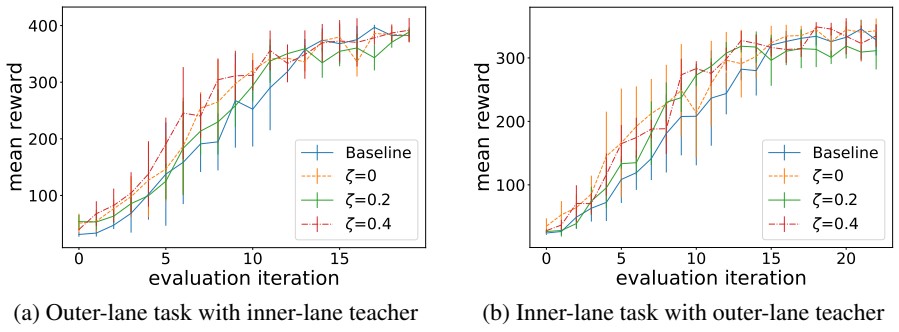

(a) Outer-lane task with inner-lane teacher          (b) Inner-lane task with outer-lane teacher

Figure 8: Evaluation performance with respect to different $\zeta$'s, averaged across five runs.

We then study the performance of training with different cross-entropy loss weights $\beta_k$. First, we fix the diminishing factor to be 0.95, namely, $\beta_{k+1} = 0.95\beta_k$, and test different $\beta_0$'s. From Figure 9, we can see that training with all $\beta_0$ values can improve the initial performance compared to the baseline. However, when the teacher task is different from the target task, larger $\beta_0$ values, like 0.3, may reduce the agent's asymptotic performance since the agent overshoots learning from teacher policy. In addition, we then fix $\beta_0 = 0.2$ and test different $\beta_k$ schedules. The results are shown in Figure 10. We can observe some trade-offs between training convergence time and final performance. By reducing the $\beta$ values faster, one can improve the final performance but increase the training time that needed to achieve some certain performance level. It is of interest to automatically determine the best $\beta_k$ values during training, which needs further investigation. We leave it as another future work.

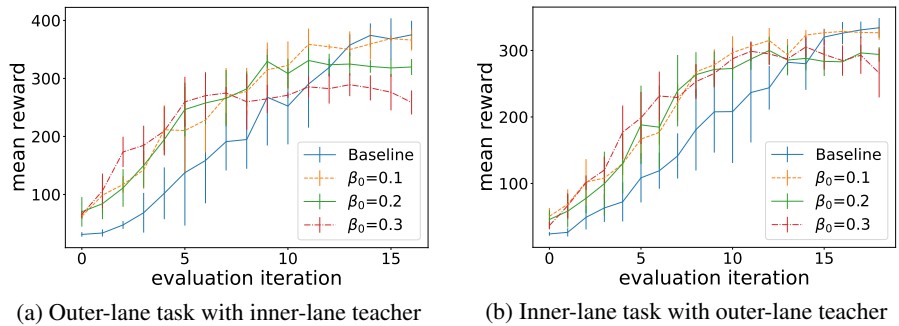

(a) Outer-lane task with inner-lane teacher

(b) Inner-lane task with outer-lane teacher

Figure 9: Evaluation performance with respect to different initial $\beta_0$'s, averaged across five runs. Here we fix the $\beta$ update to be $\beta_{k+1} = 0.95\beta_k$.

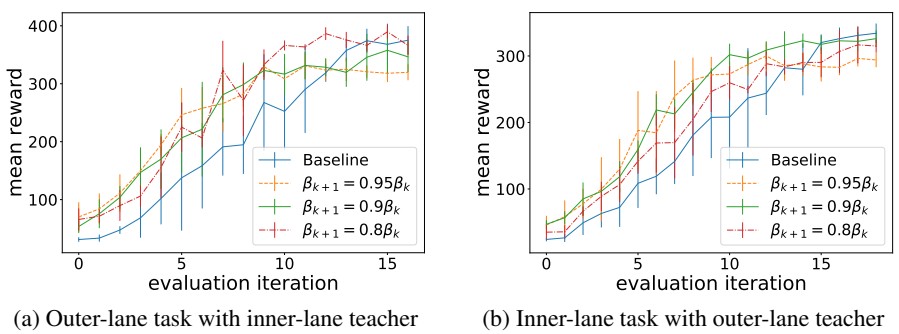

(a) Outer-lane task with inner-lane teacher

(b) Inner-lane task with outer-lane teacher

Figure 10: Evaluation performance with respect to different $\beta$ schedules, averaged across five runs.

### C.4 NEURAL NETWORK ARCHITECTURES

For completeness of the experiments, we also provide some results regarding different neural network architectures in this section. Take the DeepRacer task of multi-car racing against bot cars as an example, we have used three-layer CNN as the default architecture in experiments. Here, we present the comparison of REPAINT against other baselines with the evaluation performance using four-layer CNN (Figure 11) and five-layer CNN (Figure 12).

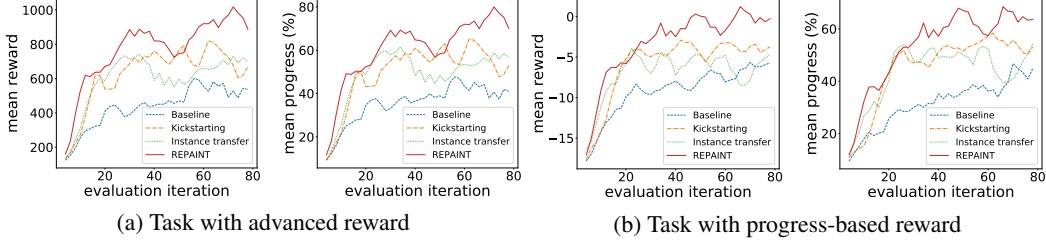

(a) Task with advanced reward

(b) Task with progress-based reward

Figure 11: Evaluation performance for DeepRacer multi-car racing against bot cars, using 4-layer CNN. The value at each iteration is smoothed by the mean value of nearest 3 iterations.

### C.5 SUMMARY OF WALL-CLOCK TRAINING TIME

In addition to the summary of reduction performance with respect to number of training iterations presented in Table 2, we also provide the data of wall-clock time in Table 5 below. Again, we can see a significant reduction by training with REPAINT, which reaches at least 60% besides the DeepRacer single-car time trial. The kickstarted training performs well when a similar teacher policy is used. Although training with only instance transfer cannot boost the initial performance, it still reduces the training cost to achieve some specified performance level.

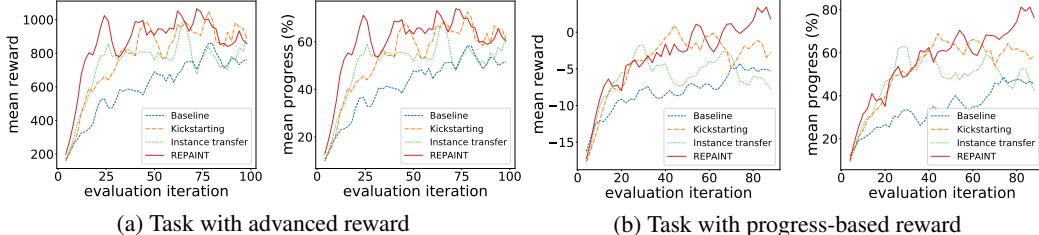

(a) Task with advanced reward             (b) Task with progress-based reward

Figure 12: Evaluation performance for DeepRacer multi-car racing against bot cars, using 5-layer CNN. The value at each iteration is smoothed by the mean value of nearest 3 iterations.

Table 5: Summary of wall-clock time of experiments.

| Env. | Training hardware | Teacher type | Target score | $T_{Baseline}$ (hrs) | $T_{KS}$ (pct. reduced) | $T_{IT}$ (pct. reduced) | $T_{REP}$ (pct. reduced) |
|---|---|---|---|---|---|---|---|
| Reacher | laptop | similar
different | -7.4 | 2.1 | 0.6 (71.4%)
0.9 (57.1%) | 1.1 (47.6%)
1.4 (33.3%) | 0.4 (81.0%)
0.6 (71.4%) |
| Ant | laptop | similar | 3685 | 19.1 | 8.0 (58.1%) | 12.8 (33.0%) | 7.5 (60.7%) |
| Single-car | AWS, p2
AWS, p2 | different
different | 394
345 | 2.2
2.3 | –
– | –
– | 1.5 (31.8%)
1.5 (34.8%) |
| Multi-car | AWS, p2
AWS, p2 | sub-task
sub-task | 1481
2.7 | 16.4
9.6 | 4.8 (70.7%)
9.3 (3.1%) | 12.6 (23.2%)
8.3 (13.5%) | 4.5 (72.6%)
3.7 (61.5%) |

