# OpenReview forum: "REPAINT: Knowledge Transfer in Deep Actor-Critic Reinforcement Learning"
_ICLR.cc/2021/Conference — Reject_

### Official Review · AnonReviewer1 · 2020-10-28
**A transfer learning method that combines kickstarting policy distillation with experience selection**

**Rating:** 4
**Confidence:** 4

**Review:**

This paper proposes a new transfer learning approach to leverage the previously learned knowledge stored in a pre-trained teacher policy to facilitate the learning in a new task. The proposed method combines an established technique termed kickstarted training with a simple experience filtering method. The kickstarted training approach augments a policy distillation loss to the actor-critic training loss. The experience filtering simply performs thresholding to filter out transitions generated by teacher policy which have reward below some threshold, where the reward for thresholding is taken from the target task.

I lean slightly towards rejection because the novelty of the method is somewhat too limited and the empirical evaluation does not give much new insights to the problem of policy transfer. The only difference between the proposed method and kickstarted training is that the method uses more data, which are the transitions generated by the teacher policy and could receive high reward at the student task domain. Such an add-on does not give much privilege in terms of return or training time.

Other detailed comments :
- The tasks for MuJoCo-Reacher are too similar. The authors only use an alternative weight on the control reward to create a distinct task. But essentially these can hardly be considered as alternative tasks for policy transfer scenario.

- I wonder how the weights for policy distillation loss are specified for the baseline method kickstarting. Do you use the same value as REPAINT for each evaluation domain?

- In MuJoCo-Reacher-similar, REPAINT and Kickstarting achieves the same standard for mean reward, and training time for the two also seem quite close to each other.

- In MuJoCo-Ant, the improvement of average return for REPAINT over Kickstarting also seem not that significant. I wonder why the results are presented in table rather than plots.

- The proposed method presents a way to quantitatively evaluate the task similarity (Eq 4.4). I'm expecting to see some finer-grained usage of such a formulation, e.g., creating tasks with positive but decreasing levels of similarity and show empirical results. However, the tasks used for the actual evaluation only have vaguely characterized similarity values, which leaves a feeling that the formulation for similarity is not necessary to have.

---

> ### Author Response · Authors · 2020-11-12
> **Author response to AnonReviewer1 with more clarification**
>
> Dear reviewer,
>
> Thanks for your efforts to provide the detailed comments. We feel sorry if you are confused about the novelty and experimental results due to our description and organization of the algorithm and experiments. We want to take this opportunity to better clarify the contributions of our submission.
>
> In the instance transfer, we proposed a new method, i.e., advantage-based experience replay, which performs thresholding to filter out transitions with $\textbf{low advantage estimates}$, but not with “low reward’’. There are several differences between REPAINT and kickstarting, which can be seen in both the algorithm and the experimental results. From the algorithm, the kickstarting method only transfers based on the representation of the source policies, but we add an instance transfer mode to it. Therefore, while the kickstarting only aims to mimic the teacher policy behavior, the instance transfer mode can selectively choose the “useful’’ samples to learn from. Our idea of filtering transitions is simple but very effective, which can be validated by our experimental results.
>
> In the experiments, we wanted to demonstrate that the proposed REPAINT algorithm can reduce the training time that needed for some certain performance level as well as improve the asymptotic performance, $\textbf{regardless of the task similarities}$. Table 2 in the last section of our submission can better summarize the results. The kickstarting method performs as well as REPAINT only when the task similarity is high, such as in MuJoCo-Reacher-similar and MuJoCo-Ant. However, when the task similarity is low, or the source task is a simple sub-task of a complex target task, e.g., the MuJoCo-Reacher-different, the DeepRacer single-car time trial, and the DeepRacer multi-car racing with progress-based reward, REPAINT outperforms kickstarting substantially. We noticed that all your detailed comments were on the experiments. Now we would like to respond to each of the comment in below.
>
> $\bullet$ $\textbf{Can different weights of the reward function create distinct tasks?}$  We believe that if each component of the reward function encodes different behavior, then changing the weights can lead to distinct tasks. For example, in the MuJoCo-Reacher task, the reward function is composed of the distance reward and the control reward. When we increase the weight of the distance reward, the agent will focus on reaching the goal point while ignoring how much movement it makes. On the other hand, when we increase the weight of the control reward, the agent tends to stay still if the goal point is hard to reach. We agree that when the two weights are close, the transfer between these two tasks is very easy. That is why we use a cosine distance function to measure the task similarity. The answer to this question is also related to your later comment on finer-grained usage of the task similarities. We will provide the response shortly.
>
> $\bullet$ $\textbf{How the weights for policy distillation loss are specified for kickstarting?}$ In order for a fair comparison, we use the same values as in REPAINT for each evaluation domain.
>
> $\bullet$ $\textbf{“The results are similar for REPAINT and Kickstarting in MuJoCo-Reacher-similar and MuJoCo-Ant’’}$. Yes. When the task similarity is high, most transfer learning methods can easily achieve high performance. We provide these results for completeness, since we want to demonstrate that REPAINT works well in all cases of task similarities, but the kickstarting method is not able to transfer well when the task similarity is low.
>
> $\bullet$ $\textbf{Why the results of MuJoCo-Ant are presented in table?}$ In the MuJoCo-Ant experiments, we noticed that the mean reward of each episode had large variance due to the fixed length of episode. Therefore, we think presenting the average return scores of some intervals can better capture the training performance curves.
>
> $\bullet$ $\textbf{Empirical results of different levels of similarity}$. Thank you for bringing that out. In the revision, we provided the empirical results for MuJoCo-Ant environment, where the task similarities are positive but in different levels. More specifically, In the teacher task, we set the weight of distance reward to be 3, 5, and 10, so that the similarity is decreasing. The new empirical results are presented in Table 1 (bottom part). From the results, we can observe that changing the reward weights can create distinct tasks, and the task similarity impacts the overall training, even when they are all related. Thanks again for the great suggestion!
>
> [To be continued]

---

> > ### Author Response · Authors · 2020-11-12
> > **Author response to AnonReviewer1 with more clarification - Continued**
> >
> > In addition, we have done some unit tests in the experiments and presented the results in the appendix, including different alternating ratios, REPAINT vs weight transfer methods, different $\zeta$ values and $\beta$ schedules, different neural network architectures, and so on. We hope these results can provide “new insights to the problem of policy transfer’’.
> >
> > Overall, we hope the above response has addressed most of your concerns, and appreciate if you have better understanding of our work and would like to re-evaluate our submission. Let us know if you have any further questions, we are happy to clarify.

---

### Official Review · AnonReviewer2 · 2020-10-28
**This paper describes an algorithm for transfer from a teacher policy that combines an existing kickstarting approach with advantage reweighted ‘instance’ transfer of the teacher.**

**Rating:** 7
**Confidence:** 4

**Review:**

Below I list the strengths and weaknesses of the paper in my opinion. Overall I vote to accept the paper for now, but my final decision will depend on the authors' clarifications to my questions below.

Strengths:
 - The algorithm is a simple combination of ideas each of which seems important according to the results.
 - Data efficiency and the ability to use previous knowledge to inform solutions to new tasks is an important problem in RL and I encourage the authors to continue to explore this space.
 - The appendix includes quite a good overview of experimental details and since the work is based on open source implementations it is potentially reproducible.
 - The appendix also includes a number of ablation studies including results on end-to-end wall clock time and additional results. As I describe below though, more can be done in terms of experimental rigor and clarification is needed for some of the presented results.

Questions/Clarifying points:
- I am concerned with the strange drops in performance in Figure 2. This instability also holds for the Baseline which is trained from scratch. This indicates it does not have to do with the transfer of the teacher policy where ostensibly the drop corresponds to critic training. Could the authors clarify what is happening here? I think it is important to have a good explanation for this.

- It is mentioned that the Beta_k parameter used in kickstarting is annealed so that it vanishes at later iterations. Could the authors clarify if this schedule is also maintained for the pure ‘kickstarting’ baseline? If it is, I’m uncertain why this baseline performs asymptotically worse on the ‘dissimilar’ tasks. Shouldn’t the influence of the teacher slowly disappear anyway?

- Figure 2b) indicates that using the top 20% metric for advantages works as well (if not better) than the one used in REPAINT. Since this metric also has the property of being scale invariant, would it not be better to use this instead?

- The authors do not really compare against any existing methods for transfer learning. For instance, it would be interesting to see how the proposed method compares to say DAGGER.

 - I’m not certain the specific task similarity measure described in the main text is a good one. For instance the ideas introduced by Carroll and Seppi and Lazaric (both referred to in the main text) attempt to quantify similarity in terms of how well an expert on the source task would perform in the target task. For instance Carroll and Seppi mention: ‘The task similarity measure should provide an approximation to the amount of learning improvement that we would get when using the source task to learn the target task under a given transfer technique’. In light of the current work, the ‘similarity’ under the proposed approach is 0 for the ‘dissimilar’ tasks. While kickstarting performs worse than REPAINT on these tasks, it still manages to do reasonably well. It would be interesting to note how dissimilar the tasks are under the other proposed metrics.

Apart from these, there are technical details which could help improve the work that did not affect my decision but list below:
- The main text mentions that the clipped loss L_clip is used even for instance transfer in Algorithm 1. Strictly speaking the importance weight used for this should be the teacher policy. This may be worth mentioning somewhere although it is not too important.
- For ease of parsing the experiments, it is often useful to have a consistent color schemes for the legend used across figures. Figure 2 and Figure 3 use the same palette but with different names attributed to each method. This isn’t a big issue but fixing it would make it easier to parse the figures and improve the overall presentation.
- The related work could also include a discussion of recent work on offline reinforcement learning. Specifically recent methods proposed by Peng et al. , Siegel et al. and Wang et al. show advantages of using different forms of the advantage function when learning from offline data. This is similar to the instance transfer prescribed in the paper where the teacher generates the fixed distribution to learn from.


References:
Siegel, N., Springenberg, J. T., Berkenkamp, F., Abdolmaleki, A., Neunert, M., Lampe, T., Hafner, R., Heess, N., and Riedmiller, M. Keep doing what worked: Behavior modelling priors for offline reinforcement learning. ICLR, 2020

Peng, X. B., Kumar, A., Zhang, G., and Levine, S. (2019). Advantage-weighted regression: Simple and scalable off-policy reinforcement learning. arXiv preprint arXiv:1910.00177.

Z. Wang, A. Novikov, K. Żołna, J. T. Springenberg, S. Reed, B. Shahriari, N. Siegel, J. Merel, C. Gulcehre, N. Heess, and N. de Freitas. Critic regularized regression. arXiv preprint arXiv:2006.15134, 2020.

S. Ross, G. Gordon, and A. Bagnell. A reduction of imitation learning and structured prediction to no-regret online learning. Journal of Machine Learning Research, 15:627– 635, 2011.


--- Edit: After rebuttal ---
I thank the authors' for their detailed response to my questions and for clarifying so many aspects of the work! I also appreciate the time taken to make the minor corrections that help with the presentation.

My main concerns have been sufficiently addressed - specifically the references to the dips in performance have eased my concerns with the validity of the experiments. The ablation with the beta parameter is also quite important I think and at least helps clarify that the issue for kickstarting may indeed be one of reaching a local optima.  Given the substantial effort (including implementing a new baseline) that went into the rebuttal I am willing to increase my score and recommend the paper be accepted at this venue.

---

> ### Author Response · Authors · 2020-11-12
> **Author response to AnonReviewer2**
>
> Dear reviewer,
>
> Thanks very much for your precise comments and helpful suggestions. In this response, we would like to answer your questions and address your concerns.
>
> $\bullet$ $\textbf{The dips in performance in Figure 2}$. Thanks for pointing that out. We have seen these dips in many experiments in our research and other papers, e.g., Figure 2 and Figure 3 in [1] (which also has the MuJoCo-Reacher experiments), Figure 13 in [2], and Figure 3 in [3]. We believe this is still an open question which requires more careful investigations. Here, we try to provide several possibilities that could cause the dips. First, as you mentioned, in the policy update, we calculate the advantage estimates based on the value network approximations, which could have large errors at the early training stage. Hence, the policy update could move towards a wrong direction. Second, exploration could be an issue. In the early stage where exploration keeps suggesting new actions, the policy network would try those actions, and thus increase the estimates of unrelated actions. Third, the problem is more likely to be task specific and algorithm dependent. It is possible that the optimization landscape in MuJoCo-Reacher environment and Clipped PPO algorithm has some distinct geometric structures.
>
> $\bullet$ $\textbf{$\beta_k$ parameter}$. Yes, we have maintained the same scheduling for the pure kickstarting baseline. In the dissimilar tasks, when the initial $\beta_0$ is relatively large and the annealing rate is relatively low, it is possible that the agent overshoots learning from the teacher policy and finds a local optimum, and thus reduce the asymptotic performance. In the revision, we provided some empirical results on the study of training with different $\beta$ values, which can be found in Appendix Section C.3 (Figure 8 and Figure 9) (now Figure 9 and Figure 10 in the latest revision). The results demonstrated that one can trade off the asymptotic performance against training convergence rate by tuning the $\beta$ scheduling. When using a smaller $\beta_0$ and faster diminishing rate, one can expect to achieve a similar final performance between kickstarting and baseline training, when the task similarity is low.
>
> $\bullet$ $\textbf{Ranking-based rule vs absolute value-based rule}$. Thanks for pointing out this great observation. We would like to state that the two rules are equivalently good. Although the ranking-based rule has the good property of being scale invariant, it takes extra effort to sort the advantages in each training episode. We added one paragraph in Section 4.3 in the revision based on your comment of scale invariance. In practice, one can consider to normalize different reward functions, so that the absolute value-based rule can also enjoy the scale invariant property.
>
> $\bullet$ $\textbf{Comparison against other transfer learning methods}$. We agree that it is interesting to compare with other transfer learning methods in RL. We have started implementing another algorithm, but we cannot guarantee to provide results by the end of discussion phases. As is also mentioned in the related work section, we want to remark that most of the transfer learning algorithms are designated to Q-learning, which are not directly comparable. We agree that the comparison with other methods could be a great plus to our paper, although we feel that the current experiments have already demonstrated the key contributions of our submission.
>
> [To be continued]

---

> > ### Author Response · Authors · 2020-11-12
> > **Author response to AnonReviewer2 - Continued**
> >
> > $\bullet$ $\textbf{Task similarity metrics}$. In our paper, we want to showcase that the REPAINT algorithm can significantly reduce the training time while still achieve high performance, regardless of the task similarities. We simply use a cosine distance function to measure the task similarity. When the cosine similarity between two tasks are 0, it means the two tasks are “orthogonal’’ to each other. Hence, we can call them “dissimilar’’ tasks. In the experiments, we always assume that the task similarity is unknown, and do not use any similarity information in the training. In contrast, the calculation of the two metrics you mentioned requires enough samples from the two MDPs, so that they can better capture the transferability. In Lazaric et al. (2008), the goal of developing the task compliance and sample relevance is to improve the transfer learning performance, when there are multiple tasks and a bunch of samples. Then it proposes an instance transfer method, which prioritizes the samples by the sample relevance. In analogy, our advantage-based filtering rule can be seen as a simpler relevance metric, since the experiments have shown that high-advantage samples from the source task are more helpful in learning the target task. In Carroll and Seppi (2005), it proposes four task similarity metrics, among which the $d_R$ function is more related to the scenario discussed in our paper, where $d_R$ is defined to be the mean squared error between the sample rewards in the source and target tasks. In continuum level, it is just the two-norm of the reward functions. However, we think the cosine distance is better than the two-norm (Euclidean) distance in our case. For example, In MuJoCo-Reacher experiment, the target task takes a reward function with a weight vector $(1,1)$, and the similar and dissimilar source tasks take weight vectors $(3,1)$ and $(-1,1)$, respectively. By two-norm, they have equi-distance to the target reward. However, the reward function with -1 in its distance reward component encodes completely different behavior for the policy, hence the corresponding source task should be viewed as a dissimilar task.
> >
> > $\bullet$ $\textbf{Regarding the technical details}$. We appreciate your suggestions on the technical details. They are all very helpful! We made revision based on your comments as follows. (1) We added one sentence in the first paragraph of Section 4.2, explicitly stating the use of teacher policy in $L_\text{clip}$. (2) We changed the colors in Figure 2 to make the legends consistent across the experiments. (3) We added one paragraph in Section 4.2 to discuss the relatedness of our advantage-based experience replay with the “advantage weighting’’ approaches in some offline RL papers.
> >
> > We hope that we did well in addressing your concerns, and we are happy to answer any further questions. In addition, we appreciate if you would like to re-evaluate our submission.
> >
> > $\textbf{References}$
> >
> > [1] Zhang, Shangtong, Wendelin Boehmer, and Shimon Whiteson. “Generalized off-policy actor-critic.’’ Advances in Neural Information Processing Systems. 2019.
> >
> > [2] Konidaris, George, Ilya Scheidwasser, and Andrew G. Barto. "Transfer in reinforcement learning via shared features." The Journal of Machine Learning Research 13.1 (2012): 1333-1371.
> >
> > [3] Barreto, André, et al. “Successor features for transfer in reinforcement learning.’’ Advances in neural information processing systems. 2017.

---

> ### Author Response · Authors · 2020-11-22
> **Comparison of REPAINT and an existing transfer learning method was provided in the revision**
>
> Dear Reviewer 2,
>
> In the latest revision, we presented some new results for the comparison against an existing method in transfer RL. In order for a fair comparison, we incorporated the kickstarting method and the relevance-based instance transfer [1], and evaluated its performance on MuJoCo-Reacher with both similar and different source tasks.
>
> From Appendix Figure 6 in the latest revision, one can see that when the source task is a similar task, both methods worked well. However, when the source task is much different from the target task, REPAINT performed significantly better than RBT. RBT failed to achieve any performance gain in this case.
>
> We hope the new results, along with the first revision, could improve your evaluation of our submission. We are happy to answer any further questions. Thank you!
>
> $\textbf{References}$
>
> [1] Lazaric, Alessandro, Marcello Restelli, and Andrea Bonarini. "Transfer of samples in batch reinforcement learning." Proceedings of the 25th international conference on Machine learning. 2008.

---

### Official Review · AnonReviewer4 · 2020-10-28
**RL domain transfer method has limited novelty**

**Rating:** 4
**Confidence:** 4

**Review:**

The proposed method relies on kickstarting, thus using policy distillation as an auxiliary loss for transferring from a source task to a target task, as a starting point. In addition, the authors add 'instance transfer', i.e., selecting some prioritized data from the source task to be used to train the target task. The combination of these two features produces fairly strong performance on various transfer experiments in four simulation environments.

The approach is clearly presented. The motivation is clear and related approaches are described. The experiments are described, although the descriptions and conclusions could be more precise.

The proposed approach has very little novelty, since kickstarting is already published and similar versions of instance transfer have been published. The instance transfer method, which is rather heuristic, requires tuning a threshold which may be a limitation. The results, moreover, do not convincingly show that putting these two features together makes a significant difference to performance. The improvements are minor for all the experiments when compared with the baseline or kickstarting.

Pros
- Clearly described method
- Well-designed experiments

Cons
- Lack of novelty beyond the combination of 2 known methods
- Results don't show that the method is substantially better than kickstarting alone

---

> ### Author Response · Authors · 2020-11-12
> **Author response to AnonReviewer4 with more clarification**
>
> Dear reviewer,
>
> Thanks for your comments and suggestions. We apologize that our description and organization of the paper have brought confusion to you, so that you may have missed some key points we want to demonstrate. If you do not mind, we would like to address each of your concerns in this response.
>
> $\bullet$ $\textbf{Novelty of the instance transfer}$. We proposed a new instance transfer framework in this work. To the best of our knowledge, there is no published "similar versions" of instance transfer algorithms. We appreciate if you could kindly refer those "similar versions" to us so that we can better assess our algorithm. Based on our literature survey, we have made explicit discussion on the related work in Section 4.2 (we also added one more paragraph in our revision). The idea of prioritizing transitions has been used in the prioritized experience replay (PER), but not in the instance transfer learning. Moreover, PER prioritizes the transitions by the TD error, and utilized importance sampling for the off-policy evaluation, while we directly filter out most of the transitions based on the advantage estimates, and perform policy update in actor-critic framework without importance sampling. In the revision, we included three more related works, which used the advantage-based prioritizing in offline RL. Again, these approaches have not been applied to instance transfer, and our proposed formulation (filtering rule) is much different from theirs. In the experiments, we have compared the performance of several prioritization rules and suggested to use the proposed filtering rule with a threshold $\zeta$ or the ranking-based filtering rule with a percentage threshold, which also has not been investigated in any existing instance transfer algorithms.
>
> $\bullet$ $\textbf{"The instance transfer method requires tuning a threshold"}$. Thanks for pointing that out. Our instance transfer method involves a threshold parameter, which requires a bit parameter tuning. However, as we have shown in the experiments, e.g., MuJoCo-Ant (the middle part of Table 1) and DeepRacer single-car time trial (Figure 7 in Appendix) (now Figure 8 in the latest revision), our algorithm is robust to the threshold parameter regardless of the task similarities. In addition, if one is concerned about the scale of the $\zeta$ parameter, he/she can either adopt the ranking based filtering rule with a percentage threshold (as discussed in the MuJoCo-Reacher experiment in Section 5.1), or normalize the reward functions (as discussed in Section 4.3 in the revision). Then one can easily reduce the training time while still achieve high performance by REPAINT.
>
> $\bullet$ $\textbf{"Results don’t show that the method is substantially better than kickstarting alone"}$. In the experiments, we wanted to demonstrate that the proposed REPAINT algorithm can reduce the training time that needed for some certain performance level as well as improve the asymptotic performance, $\textbf{regardless of the task similarities}$. We suggest to take a look at Table 2 again, which better summarizes the results. (1) When the task similarity is high, most transfer learning algorithms, including REPAINT and kickstarting, can reduce the training time and improve the asymptotic performance. The training time reduction reaches at least 60\%. In this case, REPAINT is only marginally better than kickstarting. (2) However, when the task similarity is low, REPAINT starts to outperform kickstarting substantially. For example, in the DeepRacer single-car time trial, kickstarting performs even worse than the baseline training, while REPAINT can still reduce the training time by around 30\% without losing the reward performance. In another case, REPAINT also significantly outperforms kickstarting in MuJoCo-Reacher-different (see the second row of Table 2 or the right side of Figure 2(a)). (3) In addition, when the source task is a sub-task of the target task, kickstarting does not always perform well (see the last two rows of Table 2, especially the last row), while REPAINT consistently achieves high performance with much less training time. More results that showcase the superiority of REPAINT over kickstarting can be found at Sections C.4 and C.5 in the appendix.
>
> We sincerely hope our response can help address your concerns on the novelty and experiments, and re-evaluate our submission with better understanding. We are more than happy to answer any further questions. Thanks!

---

### Official Review · AnonReviewer3 · 2020-10-29
**A well written paper suggesting a hybrid approach for transfer learning**

**Rating:** 6
**Confidence:** 4

**Review:**

This paper deals with transfer learning in RL. The problem broadly defined is to improve performance of an agent on a new task, given an agent (teacher) trained on a previous (different) task. There are multiple approaches to this from fine-tuning, distillation to instance transfer. A central question in this field is how to transfer learned useful behavior even when task similarity is low. While most methods would work well on similar tasks, naively applying, for example distillation, would hurt when training on a very different target task.

This paper suggests an approach that builds on two approaches: kickstarting and instance transfer. Kickstarting can be viewed as distillation of a policy with a dynamically tuned coefficient that control how much to weigh the distillation loss against the actor-critic loss. Intuitively this distillation is more helpful at the start of training and the original work uses population based training to automatically anneal this. Applying this as-is to tasks with very different reward functions would lead to a degradation in final performance on the new task.

The second approach considered here is instance transfer. Briefly, we consider trajectories from the teacher and train the agent on those that are "good" as filtered by the advantage under the current reward function.

The results are encouraging, especially showing that performance is not lost when tasks are dissimilar.  The writing overall is very clear. Below are few specific notes/questions.

* One suggestion if for greater discussion into why this works. It was not clear to me on reading what the fundamental rationale for combining the approaches in this way is, as opposed to say combining kickstarting with some other method like successor features.

* How sensitive is the method to the schedule for kickstarting? Could a well hand-tuned kickstarting schedule alone achieve all of the performance gain if we knew a priori how similar the tasks were?

* The paper mentions automatic identification of task similarity as future work -- do the authors have ideas on how best to do this?

Overall I found this a well-written paper that combines two existing approaches to transfer learning in RL to provide a unified algorithm that can do well in settings with low or high task similarity.  I think the results are promising although given the size of the error bars on some of the plots it's not clear it is completely worth the effort to use this method in all cases. Addressing the main point above on why this is necessarily a principled approach as opposed to just an ensemble of methods would help motivate this. Nonetheless in its current form I would recommend an accept.

---

> ### Author Response · Authors · 2020-11-12
> **Author response to AnonReviewer3**
>
> Dear reviewer,
>
> Thank you so much for your careful reading through our submission! Your summary has very well captured the message we wanted to deliver. There is only one thing we want to remark that the instance transfer with advantage-based experience replay is not an "existing approach", although the idea of advantage prioritization has been used in some other RL topics. Moreover, our formulation of advantage filtering is different from other works. We have made an explicit discussion in Section 4.2 (we also added one more paragraph in our revision).
>
> Based on your questions, we would like to provide our response as follows.
>
> $\bullet$ $\textbf{REPAINT vs other hybrid or ensemble methods}$. First, alternately applying kickstarting and advantage-based instance transfer is only a variant of the proposed REPAINT algorithm. We also provided an integrated version of REPAINT in Algorithm 2 of Appendix A and involved more general scenarios. By tuning the parameters $\alpha_1$ and $\alpha_2$, Algorithm 2 can be viewed as an "ensemble version" of the two transfer modes. Second, we think it is interesting to compare REPAINT against combining kickstarting with other instance transfer approaches. We have started implementing another algorithm, but we cannot guarantee to provide some results by the end of discussion phases. This could be a great plus to our work, although we think the current experiments have already demonstrated our key contributions. Moreover, we want to remark that the successor features approach will not be an option for now, since there is no open-sourced example code for reproducibility.
>
> $\bullet$ $\textbf{Sensitivity to the schedule for kickstarting}$. In the revision, we assessed the performance of kickstarting with different $\beta$ scheduling in DeepRacer single-car time-trial environment, which can be found in Appendix Section C.3, especially Figure 8 (different $\beta_0$’s with fixed schedule) and Figure 9 (different schedules with fixed $\beta_0$). (The two figures are now Figure 9 and Figure 10 in the latest revision). Our observation is that, when the task similarity is low, one can trade off the training convergence rate against the final performance. Therefore, when the task similarity is known a priori, one can hand-tune the $\beta_k$ values for better performance. In comparison, REPAINT can achieve better performance with less training time more easily, in all cases of task similarities.
>
> $\bullet$ $\textbf{Future work}$. We want to define a new metric to automatically measure the similarity of two MDPs. A good similarity metric should be highly correlated to the transferability of a task or some samples. Moreover, the computational cost for such a metric should not be too expensive, namely, the similarity can be calculated based on few samples from two MDPs. There are some metrics that have been developed in existing literature. For example, [1] defines the task compliance and the sample relevance, and proposes an instance transfer algorithm based on that. [2] measures the distance between two MDPs using a Restricted Boltzmann Machine. [3] proposes a method to measure the states’ distance in different MDPs based on the bisimulation metric. There are also many similarity metrics developed for supervised and unsupervised learning. Once a good similarity metric is given, we then aim to develop an algorithm to better automatically specify a good range of $\beta$ and $\zeta$ parameters, with less or no hand tuning.
>
> We hope our response well answered your questions, and appreciate if you would re-evaluate our submission.
>
> $\textbf{References}$
>
> [1] Lazaric, Alessandro, Marcello Restelli, and Andrea Bonarini. "Transfer of samples in batch reinforcement learning." Proceedings of the 25th international conference on Machine learning. 2008.
>
> [2] Ammar, Haitham Bou, et al. "An automated measure of mdp similarity for transfer in reinforcement learning." Workshops at the Twenty-Eighth AAAI Conference on Artificial Intelligence. 2014.
>
> [3] Song, Jinhua, et al. "Measuring the distance between finite Markov decision processes." Proceedings of the 2016 international conference on autonomous agents & multiagent systems. 2016.

---

> ### Author Response · Authors · 2020-11-22
> **More results are provided in the latest revision**
>
> Dear Reviewer 3,
>
> Based on your suggestion, we further compared our REPAINT algorithm against a method of combining kickstarting with the relevance-based instance transfer ([1]). From Appendix Figure 6 in the latest revision, one can see that when the task similarity is low, our REPAINT algorithm performed significantly better than RBT.
>
> We hope the new results can further address your concern. If so, we appreciate if you could consider to re-evaluate the overall rating. Let us know if you have any other questions.
>
> $\textbf{References}$
>
> [1] Lazaric, Alessandro, Marcello Restelli, and Andrea Bonarini. "Transfer of samples in batch reinforcement learning." Proceedings of the 25th international conference on Machine learning. 2008.

---

> > ### Comment · AnonReviewer3 · 2020-11-23
> > **Re**
> >
> > Thank you for your response!

---

### Author Response · Authors · 2020-11-12
**Revision Uploaded**

In the revision, we have made the following changes:

$\bullet$ We added a paragraph in Section 4.2 to discuss the relatedness of our advantage-based prioritization and the "advantage weighting" approach that has been used in some papers of offline RL.

$\bullet$ We added a paragraph in Section 4.3 to discuss the $\beta$ and $\zeta$ parameters. We stated that although they are task specific, one can consider to normalize the reward functions in practice, to keep them in the same scale.

$\bullet$ We changed the line colors in Figure 2 to make the legends consistent across all the experiments.

$\bullet$ We added more experimental results for MuJoCo-Ant in Table 2, where we investigated the transfer performance of different teacher policies with different levels of task similarity.

$\bullet$ In Appendix Section C.3, we added more experimental results for DeepRacer single-car time trial, where we further investigated the $\beta_k$ schedules with fixed $\beta_0$. The new results are shown in Figure 9 (now Figure 10 in the latest revision).

We thank all the anonymous reviewers for their valuable suggestions!

---

### Author Response · Authors · 2020-11-12
**More clarification on the novelty and experimental results - Part 1**

We thank all the anonymous reviewers again for their useful comments. Among the reviews, we noticed that the common concerns against a clear acceptance were about the novelty and the experimental results. Therefore, we would like to provide more clarification in this thread on these two points.

Cited from [1], the goal of a transfer reinforcement learning agent varies as follows: (1) The initial performance of an agent may be improved by learning from a pre-trained policy, (2) the final performance and total accumulated reward of the agent after training may be improved via transfer, (3) the training convergence time or the learning time needed by the agent to achieve a specified performance level may be reduced via knowledge transfer.

Motivated by the above three goals, we have proposed the REPAINT algorithm which incorporates an existing representation transfer approach (the kickstarting) with a newly developed instance transfer method. The kickstarting method has been shown to be effective in boosting the agent’s initial performance, which is exactly the first goal we want to achieve. However, when the task similarity between the source and target tasks is low, kickstarting cannot always achieve the other two goals (which has been validated in our experiments). To this end, we thus proposed an instance transfer method with advantage-based experience replay. From the experiments, we can observe that although the instance transfer method alone works well in both similar and different tasks, the improvement is not significant. Therefore, we ended up with an integrated algorithm REPAINT and presented it in this paper.

In below we will discuss the novelty and experiments in greater detail.

$\bullet$ $\textbf{Novelty}$

Motivated by the aforementioned three goals, we proposed the REPAINT algorithm and provided two variants, namely, the alternating version (Algorithm 1) and the integrated version (Algorithm 2 in Appendix Section A). Moreover, we claimed that the REPAINT algorithm is well generalized, which can be easily extended to other policy gradient based algorithms. Please see Appendix Section A for more discussion.

Our major novelty leans on the development of the advantage-based experience replay, by which the agent can select “good” teacher samples and ignore “bad” samples. As a result, the agent can always transfer “useful” knowledge from the teacher task with any cases of similarity. The idea of prioritizing and re-weighting samples has been widely used in reinforcement learning. However, to the best of our knowledge, we are the first to use a threshold advantage-filtering in the instance transfer. In Section 4.2, we made an explicit discussion on the related works that had applied the similar idea. We also made an explicit comparison in the experiments of our proposed filtering rules against the well-known prioritized experience replay method (see Figure 2(b)). In addition, we also connected our formulation with the off-policy actor-critic, and discussed how the proposed sampling experience impacted the policy gradient update. Based on that, we made some suggestions on how to better use the REPAINT algorithm in practice.

---

> ### Author Response · Authors · 2020-11-12
> **More clarification on the novelty and experimental results - Part 2**
>
> $\bullet$ $\textbf{Experiments}$
>
> From the reviewers’ comments, we noticed that the reviewers paid special attention to the final return scores of each model. We want to emphasize that the performance of transfer learning should be evaluated in three metrics, which corresponds to the above three goals: (1) the initial performance, (2) the final return scores (or the asymptotic return scores), and (3) the training time to a return threshold. The final return score is not the only metric and sometimes is not the most important in transfer reinforcement learning. Our figures and tables can reflect all three metrics for each model. Moreover, Table 2 in the last section better quantized and summarized our experimental results, especially for the metrics of time to threshold and final return score.
>
> We believe that our experimental results in the main text have demonstrated the following points: (1) When the task similarity between source and target tasks is high, most transfer RL algorithms perform well compared to the baseline training. The algorithms include REPAINT, kickstarting, the proposed instance transfer method, and the direct weight transfer method (warm-start). In particular, REPAINT and kickstarting are the best. (2) However, when the task similarity is low, the performance of training with kickstarting starts to degrade. Although it can still jump start the training, the final return scores and the time to score threshold are not always improved. A notable example is the DeepRacer single-car time trial, where the kickstarting never achieved the same performance level as the baseline training. In another example, i.e., the MuJoCo-Reacher, where both similar and different tasks were considered, we can see a clear performance degradation for kickstarting between the two. In comparison, the REPAINT can still reduce the training time by 30\% in the DeepRacer single-car environment without losing the final return scores. Moreover, it performed consistently well in the two MuJoCo-Reacher tasks. (3) When the teacher task can be viewed as a sub-task of the target task, REPAINT still performed consistently well, while the transfer performance of kickstarting highly relied on the task complexity and the reward function. This can be observed in the DeepRacer multi-car racing experiments.
>
> In the appendix, We also provided more extensive results about the unit tests and ablation study, including the alternating ratios of the two transfer modes, the comparison of REPAINT against the weight transfer method (warm-start), the effects of using different $\zeta$ values and $\beta$ schedules, and the effects of using different neural network architectures. Besides, we also provided the source code and the detail of hyper-parameters we have used in the experiments, in order for the reproducibility. We believe our experiments have great completeness and successfully tell a compelling story.
>
> Overall, this thread summarizes the motivation of the paper, our novelty of the algorithm, and the messages one can get from the experiments. We hope the reviewers would kindly re-evaluate our submission if we have addressed their concerns.
>
> $\textbf{References}$
>
> [1] Taylor, Matthew E., and Peter Stone. “Transfer learning for reinforcement learning domains: A survey.” Journal of Machine Learning Research 10.Jul (2009): 1633-1685.

---

### Author Response · Authors · 2020-11-22
**New Revision Uploaded**

We added more experimental results in this revision. As suggested by Reviewer 3 (also mentioned by Reviewer 2), we compare REPAINT with another method, namely, combining the kickstarting with the relevance-based transfer proposed in [1]. The results can be found in Appendix Section C.1 (more specifically, Figure 6).

From the results, we can again observe that when the source task is similar to the target task, both methods worked well. However, when a dissimilar target task is used, RBT failed to achieve any performance gain from instance transfer. The performance of REPAINT is significantly better than RBT in this case.

We hope the new results have further shown our contributions in the knowledge transfer for actor-critic RL.

$\textbf{References}$

[1] Lazaric, Alessandro, Marcello Restelli, and Andrea Bonarini. "Transfer of samples in batch reinforcement learning." Proceedings of the 25th international conference on Machine learning. 2008.

---

### Decision · Program_Chairs · 2021-01-07
**Final Decision**

**Decision:**

Reject

**Comment:**

The is a borderline paper with the reviewers split in their recommendations.  The decision is therefore not easy.

The work is promising, but a key concern is that the contribution appears incremental: the paper proposes to alternate between kickstarting, which is itself not entirely new as an idea, with a simple instance transfer heuristic.  The resulting method is straightforward, which can be considered a strength, but there is no serious technical justification beyond intuitive motivation.  Rather than present technical analysis, the paper focuses more on intuitively delivering the proposal then evaluating it.  This would be acceptable if the empirical outcomes were undeniably impressive, but the outcomes, though positive, are not overwhelming.  The experimental evaluation is limited in scope, considering only the simplest MuJoCo environments and a benchmark racing simulator.

The authors responded to some of the criticisms forcefully, and were comprehensive in their rebuttal, but if the support for the proposed method is to be entirely intuitive and empirical, one would have expected a more comprehensive evaluation where transfer was used to solve more impressively difficult problems.  Overall, I think this work would be better served by adding a technical analysis that validates the significance of the instance transfer heuristic, combined with a broader empirical study that tackles more challenging domains.